# Central Sensitization in Cancer Survivors and Its Clinical Implications: State of the Art

**DOI:** 10.3390/jcm12144606

**Published:** 2023-07-11

**Authors:** Tomohiko Nishigami, Masahiro Manfuku, Astrid Lahousse

**Affiliations:** 1Department of Physical Therapy, Faculty of Health and Welfare, Prefectural University of Hiroshima, Hiroshima 723-0053, Japan; 2Graduate School of Comprehensive Scientific Research, Prefectural University of Hiroshima, Hiroshima 723-0053, Japan; sunday_attinente@yahoo.co.jp; 3Department of Rehabilitation, Breast Care Sensyu Clinic, Osaka 596-0076, Japan; 4Pain in Motion Research Group (PAIN), Department of Physiotherapy, Human Physiology and Anatomy, Faculty of Physical Education & Physiotherapy (KIMA), Vrije Universiteit Brussel, 1090 Brussels, Belgium; astrid.lucie.lahousse@vub.be; 5Chronic Pain Rehabilitation, Department of Physical Medicine and Physiotherapy, University Hospital Brussels, 1090 Brussels, Belgium; 6Research Foundation Flanders (FWO), 1000 Brussels, Belgium; 7Rehabilitation Research (RERE) Research Group, Department of Physiotherapy, Human Physiology and Anatomy, Faculty of Physical Education & Physiotherapy (KIMA), Vrije Universiteit Brussel, 1090 Brussels, Belgium

**Keywords:** cancer survivors, central sensitization, central sensitivity syndrome, insomnia, stress, pain neuroscience education, precision medicine

## Abstract

Although the prevalence of cancer pain is 47% after treatment, cancer pain is often underestimated, and many patients are undertreated. The complexity of cancer pain contributes to the lack of its management. Recently, as the mechanism of cancer pain, it has become clear that central sensitization (CS) influences chronic pain conditions and the transition from acute to chronic pain. In this state-of-the-art review, we summarized the association of CS or central sensitivity syndrome with pain and the treatment for pain targeting CS in cancer survivors. The management of patients with CS should not only focus on tissue damage in either the affected body regions or within the central nervous system; rather, it should aim to target the underlying factors that sustain the CS process. Pain neuroscience education (PNE) is gaining popularity for managing chronic musculoskeletal pain and could be effective for pain and CS in breast cancer survivors. However, there is a study that did not demonstrate significant improvements after PNE, so further research is needed. Precision medicine involves the classification of patients into subgroups based on a multifaceted evaluation of disease and the implementation of treatment tailored to the characteristics of each patient, which may play a central role in the treatment of CS.

## 1. Introduction

In high-income nations, cancer now ranks above vascular illnesses as the main cause of mortality [1]. Additionally, it is anticipated that by 2040, the worldwide cancer burden will increase by 47% [2]. Oncology has made significant progress, and advanced cancer is no longer synonymous with terminal illness. However, providing pain treatment during the survivorship phase is gaining more importance due to the expanding population of cancer survivors [2]. The prevalence of cancer pain is 47% (95%CI 39–55) after treatment [3]. Despite this high prevalence, cancer pain is often underestimated, and many patients are undertreated [4,5,6,7]. The complexity of factors affecting cancer pain is contributed to the lack of management of cancer pain [8,9,10]. Pain in cancer survivors can be difficult to manage because they underwent many types of treatment, including surgery, radiation therapy, and chemotherapy, and clinicians should be cautious because the pain might be due to cancer metastasis/recurrence or other non-cancer-related causes. Therefore, to improve the lack of management of cancer pain, the International Association for the Study of Pain (IASP) defined a new classification of cancer survivor pain in ICD-11 [11]. The new classification of cancer-related pain in cancer survivors is divided into two major categories: “chronic cancer pain”, such as visceral pain and bone metastasis pain due to cancer progression or metastasis, and “chronic pain after cancer treatment” related to surgery or drug treatment [11]. More recently, patients with chronic musculoskeletal pain have been classified into three pain mechanisms: “Nociceptive pain”, “Neuropathic pain”, and “Nociplastic pain” as a classification of pain properties [12,13] and cancer pain is classified in the same way [14]. Clinicians should consider the seven-step diagnostic approach to differentiate between predominant pain and provide appropriate pain treatment in cancer survivors [14]. 

Recently, as the mechanism of cancer pain, it has become clear that central sensitization (CS) influences chronic pain conditions and the transition from acute to chronic pain [15,16,17,18]. IASP defines CS as the “Increased responsiveness of nociceptive neurons in the central nervous system to their normal or subthreshold afferent input”. Systematic reviews and meta-analyses of CS for musculoskeletal diseases have reported that CS influences symptom severity and pain in musculoskeletal diseases such as knee osteoarthritis and low back pain [18,19,20,21,22]. CS has also received attention as a mechanism for cancer pain because CS could affect pain in about 40% of breast cancer survivors [23,24]. Moreover, CS-related symptoms have the capability to predict the intensity and interference of persistent post-surgical pain 1 year after surgery [25,26]. These findings suggest that the assessment and treatment of CS are important for the appropriate treatment and management of cancer pain.

## 2. Objectives

This state-of-the-art review aims to investigate the relationship between CS or central sensitivity syndrome and pain in cancer survivors, as well as explore the treatment approaches targeting CS for pain management. While CS is initially observed in animal models, this review focuses on its assumed presence in humans. Additionally, the review will elaborate on the potential associations with other comorbidities that may contribute to the perpetuation of CS in cancer survivors. Furthermore, it will provide insights and directions for future research, along with discussing the clinical implications of CS in the context of pain management for cancer survivors. 

## 3. Methodology

A comprehensive search was conducted on PubMed and Web of Science until April 2023, using keywords such as Cancer Survivors, Central Sensitization, Central sensitivity syndrome, Central Sensitization-related symptoms, Insomnia, Sleep Disturbances, Stress, Pain Neuroscience Education, and Precision Medicine in order to identify the most relevant and up-to-date evidence. Eligible articles must meet the following requirements: (1) be written in English, Dutch, French or Japanese, (2) be published in full text, and (3) be consistent with the goal of this review. The following study designs were not included in the studies: case reports, conference proceedings, abstracts, letters to the editor, statements of personal opinion, and editorials. T.N. and M.M. conducted the initial literature review, and all co-authors subsequently contributed to revisions and additions. The original draft of the text was written by T.N. and M.M., and all authors engaged in electronic communication to discuss and revise the final draft. With reference to the classification of cancer survivor pain in ICD-11 [11], cancer pain is caused by damage of primary cancer, metastasis (e.g., bone pain or visceral metastasis pain), or cancer treatments, and these treatments can induce chronic secondary pain syndromes that persist after cancer treatment such as postmastectomy pain or post-thoracotomy pain after surgery, chemotherapy-induced peripheral neuropathy (CIPN), aromatase inhibitor-induced musculoskeletal symptoms, radiation-induced neuropathy or radiation-induced fibrosis. This paper distinguishes (1) pain related to cancer or during its treatment and (2) persistent pain after treatment completion (except for maintenance therapy).

## 4. Pain Related to Central Sensitization in Cancer Survivors

Assessing CS in individuals remains a challenge, and an optimal clinical approach for this purpose is quantitative sensory testing (QST). QST utilizes standardized mechanical and thermal stimuli, such as von Frey filament pinprick stimuli, light touch, pressure algometers, and quantitative thermosetting, to explore the nociceptive and non-nociceptive afferent pathways in the peripheral and central nervous systems. There are two main modalities of QST: static QST and dynamic QST. Static QST is the most basic method of evaluating response to standardized stimuli. Pressure value at the moment when the pressure stimulus changes to pain is called the pressure pain threshold (PPT) and is one of the most frequently used static QST. Dynamic QST is an evaluation that reflects functional changes in the central pain regulatory system and requires a slightly more complicated method than static QST. Temporal summation of pain (TS) examines the phenomenon of pain exacerbated over time by continuous pain stimulation. Conditioned pain modulation (CPM) is the suppression of pain sensitivity at the site of evaluation by a pain stimulus applied to a remote site. TS evaluates the hyperresponsiveness of the ascending pain control system, while CPM evaluates the dysfunction of the descending inhibitory controls. The utility of QST for analyzing the various etiologies and pathologies in musculoskeletal pain disorders is evident [27]. For clinicians, there is growing interest in bedside QSTs, which do not necessitate specialized, expensive, or time-consuming equipment [28,29]. Additionally, the validity and reliability of bedside QSTs are promising [30,31,32]. However, future research needs to assess its added value and feasibility in clinical practice for assessing CS [33].

There is also an increasing number of reports related to hyperalgesia and CS measured using QST [25,26,34,35,36,37,38,39,40,41]. Most reports on cancer pain and QST have been mainly evaluated by PPT. Postoperative breast, head and neck, and colorectal cancer survivors have more hypersensitivity in the surgical neck, shoulder joint, and lumbar back compared to healthy controls [37,38,39]. Several studies have revealed that hypersensitivity has been observed in distant areas from the painful site, such as the nonoperative neck and shoulder joint [37,38] and the tibialis anterior muscle as well as the operative side and in painful areas [37,40,41]. Survivors with chronic postoperative pain after breast cancer surgery have decreased CPM and enhanced TS compared to survivors without chronic pain [42,43]. Edwards RR et al. reported that pain catastrophizing might mediate central nervous system pain-modulatory processes [43]. Scott et al. reported that radiotherapy for bone metastatic pain improves hypersensitivity at the pain site [44]. However, there are no reports on hypersensitivity at distant areas from the pain site, and the relationship between cancer pain (e.g., bone metastatic pain and visceral metastatic pain) and CS is not clear. Thus, further research is needed to determine whether the relationship between post-cancer treatment pain and CS in cancer survivors is similar for cancer pain.

## 5. Pain Related to Central Sensitivity Syndrome in Cancer Survivors

Yunus et al. proposed the central sensitivity syndrome as a comprehensive disease concept in which CS is involved in pathogenesis [45]. Unexplained organic symptoms related to CS common to various chronic diseases consider symptoms a single syndrome rather than in isolation. This terminology is a breakthrough that corrects the idea that different diagnoses have different mechanisms. Recently, the Central Sensitization Inventory (CSI) was proposed as an alternative method and a comprehensive screening tool for evaluating CS-related symptoms. CSI consists of symptoms associated with worsening CS, such as sleep disturbances, muscle stiffness, fatigue, sensitivity to light and smell, and stress (CS-related symptoms), and has been translated in many countries [46]. The CSI has shown excellent psychometric properties in chronic pain patients with CS-related symptoms [46] and excellent validity and internal consistency in breast cancer patients [47]. Higher CSI scores indicate a higher degree of self-reported CS-related symptoms, which can be classified into three clusters of severity: low level, medium level, and high level [48].

CS-related symptoms contribute to the prevalence of chronic pain after breast cancer surgery, pain intensity, and capacity impairment [23,35,49,50,51]. It has also been found that breast cancer survivors with medium and high levels of CS severity have more pain intensity and pain location than breast cancer survivors with low levels of CS severity [47]. In a longitudinal study, CS-related symptoms before and after surgery were independent predictors for pain intensity and disability of chronic pain after breast cancer surgery, in addition to treatment-related factors such as axillary lymph node dissection [25,26]. CS-related symptoms are not only associated with pain intensity and disability but also with anxiety, depression, pain catastrophizing [52,53], and fear of exercise [54]. The association between pain after cancer treatment and CS-related symptoms in cancer survivors is clear, but the association with cancer pain (e.g., bone and visceral metastases) is still unclear. However, cancer survivors with advanced cancer pain and those receiving palliative care or opioid treatment generally have more CS-related symptoms, such as insomnia and fatigue [55,56,57,58]. Thus, assessment and intervention for CS-related symptoms will be important for cancer survivors with cancer pain (e.g., bone and visceral metastases) in the future as part of cancer pain management.

## 6. Inflammation and Central Sensitization in Cancer Survivors

Inflammation has been shown to play a role in both the initiation and persistence of central sensitization [38,59]. Under normal conditions, astrocytes and microglia are primarily responsible for maintaining cell retention and immune responses in the spinal cord. However, when inflammation occurs, these cells become activated. For instance, activated astrocytes release inflammatory cytokines like Interleukin-1 beta (Il-1β) and Tumor Necrosis Factor-alpha (TNF-α), which contribute to the development of central sensitization [60]. Similarly, the activation of microglia leads to the release of inflammatory cytokines, Prostaglandin E2 (PGE2), Nitric Oxide (NO), and Brain-Derived Neurotrophic Factor (BDNF). Notably, BDNF can suppress the function of inhibitory Gamma-Aminobutyric Acid (GABA)-ergic neurons that densely reside in layer II of the dorsal horn of the spinal cord [61]. These mechanisms collectively contribute to the heightened excitability of spinal dorsal horn neurons and the occurrence of central sensitization. Moreover, microglia play a significant role in maintaining advanced-stage cancer pain in female rats by generating the inflammatory cytokine IL-1β and increasing the synaptic transmission of spinal nociceptive neurons [62]. Despite the likelihood of inflammation’s involvement in CS among cancer survivors, there is currently only the support of preclinical experiments, and there is a lack of studies evaluating inflammatory markers in this population and investigating their association with CS. This remains an important area for future research, which will enhance our understanding of how to tackle inflammation for cancer pain and post-treatment pain in this cancer survivor population. 

## 7. Sleep/Insomnia and Stress Related to Central Sensitization in Cancer Survivors

Sleep disturbances [63,64] and stress [65] are common comorbidities in cancer survivors, and both are associated with a worsening of pain symptoms in chronic pain patients [66,67]. Furthermore, CS-related symptoms measured using the CSI are strongly associated with sleep disturbances and stress [68,69].

For sleep, the evidence demonstrated that taking medications/opioids disrupts individuals’ sleep quality since it might amplify their daytime fatigue and sleepiness, leading to napping during the day and disturbing the night rest [70]. Systematic review and meta-analysis suggested that sleep deprivation exacerbated peripheral and central pain sensitization measured using the QST in healthy individuals. However, similar results in cancer survivors with persistent pain remain unknown [71]. Furthermore, Pacho-Hernández JC et al. reported that sleep quality mediated CS-related symptoms and quality of life in individuals with post-COVID-19 pain [72]. 

Stress in patients with chronic pain can modulate pain and exacerbate symptoms (such as fatigue and cognitive impairments) in response to stress [73,74]. Stress and pain demonstrate a high degree of comorbidity, indicating a considerable overlap in both conceptual and biological mechanisms [67].

However, the relationship between CS and sleep quality and stress, and whether sleep quality and stress mediate for CS, is currently unclear in cancer survivors. Thus, evidence is still scarce, but it is a potential target for treating CS.

## 8. Nociplastic Pain Related to Central Sensitization in Cancer Survivor

CS is one of the key mechanisms of nociplastic pain. Nociplastic pain was proposed as a third mechanistic pain descriptor in addition to nociceptive and neuropathic pain by IASP in 2017 [12]. The IASP defines nociplastic pain as “pain that arises from altered nociception despite no clear evidence of actual or threatened tissue damage causing the activation of peripheral nociceptors or evidence for disease or lesion of the somatosensory system causing the pain” [13]. The Cancer Pain Phenotyping (CANPPHE) Network reports that the grading system guideline consists of seven steps, all of which are recommended to be implemented for cancer survivors [14]. The evaluation used in CS and CS-related symptoms is also used in the guideline (e.g., evoked pain hypersensitivity phenomena, history of pain hypersensitivity, comorbidities associated with hyperalgesia). In the future, it may become more common to use these guidelines to classify and identify phenotypes rather than to evaluate only CS or CS-related symptoms. However, at present, the reliability and validity of the guideline in cancer survivors are not clear, and further research is crucial. 

## 9. Challenges of Treating Pain in Cancer Survivors—Targeting Central Sensitization

Pharmacological treatment (NSAIDs, antidepressants, anticonvulsants, opioids, etc.) and non-pharmacological treatment (rehabilitation, cognitive behavioral interventions, etc.) are generally recommended in guidelines [8,9,10] for cancer pain. Pharmacological treatment is only a part of cancer pain management due to its numerous side effects. The effectiveness of the pharmacological treatment is also generally limited in patients with chronic non-cancer pain and CS. The use of opioids is not recommended for nociplastic pain involving CS [75,76]. According to the literature, opioids can lead to opiate-induced hyperalgesia, which will generate more pain in the long term and might decrease the survival rate [77,78].

What treatment is needed for CS? The management of patients with CS should not only focus on tissue damage (scar formation, muscle shortening, nerve damage, metastatic bone tumors, etc.) in either the affected body regions or within the central nervous system; rather, it should aim to target the underlying factors, including illness beliefs, stress, poor sleep, physical (in)activity, and even potentially unhealthy dietary habits, that sustain the CS process [79]. A systematic review revealed that physical therapy such as manual therapy, exercise, electrotherapy, education, and acupuncture improved CS-related variables in patients with chronic musculoskeletal pain [80]. A systematic review revealed that physical therapy results in a modest improvement in CS variables such as TS and CPM in patients with chronic musculoskeletal pain. It is not clear whether physical therapy improves CS variables in patients with cancer pain and pain after cancer treatment because the systematic review did not include them.

In the field of oncology, there have been attempts to see if these rehabilitations are effective [81,82,83,84,85]. International multidisciplinary roundtable reported consensus exercise guidelines [82]. The data were deemed sufficient to suggest exercise for several cancer-related health outcomes (such as fatigue, sadness, anxiety, and lymphedema). However, due to the lack of evidence, exercise for cancer pain management was not included [86]. As with other management methods, pain education is getting a lot of attention. Pain neuroscience education (PNE), an educational intervention, is gaining popularity for managing chronic musculoskeletal pain. The goal is to change the perception of pain from being caused by biological processes such as tissue damage or disease to being a necessary response to protect the body’s tissues. There are some differences between PNE for musculoskeletal pain and cancer pain PNE (Table 1). In particular, the description of the anxiety and threat of cancer recurrence is characteristic [87,88,89,90,91]. PNE alone is not effective enough, and its benefits can increase when combined with exercise. Several systematic reviews and meta-analyses have reported that interventions combining PNE and exercise therapy for persons with chronic musculoskeletal pain have resulted in at least short-term improvements in pain and disability [91]. We reported that pain intensity and disability significantly improved, and CS-related symptoms decreased in the group that received PNE combined with physiotherapy rather than the group that received biomedical education (BME) combined with physiotherapy in a retrospective case–control study of postoperative breast cancer survivors [88]. A single-arm study in breast cancer survivors suggests that the combination of exercise therapy and educational programs improves CS-related symptoms [54], and personalized eHealth interventions, including pain science education and self-management strategies, are effective in improving pain-related function, CS-related symptoms and quality of life [89]. However, in a large randomized controlled trial (RCT) of breast cancer survivors, there were no significant differences in pain-related disability, pain intensity, or psychological symptoms between the BME plus physical therapy and PNE plus physical therapy groups [90]. The results may have been influenced by the diversity of patients, including postoperative pain, CIPN, and hormone-induced arthralgia. A systematic review including more than 4000 participants found that compared to the target group, pain education programs for cancer survivors with cancer pain showed significant improvements in pain intensity and disability, self-efficacy, pain knowledge and barriers, and medication adherence, but in less than 20% of all eligible patients [92]. Combining physical therapy with a pain education program as a non-pharmacological treatment for cancer pain with cancer survivors may effectively improve pain intensity, capacity impairment, and CS-related symptoms. However, since most intervention studies have been conducted in breast cancer survivors, it is unclear whether similar results can be obtained in other cancer survivors. Further research is also needed to determine whether the pain education program is effective for all types of cancer pain, including chronic pain after cancer treatment (postoperative pain, CIPN, etc.) and chronic cancer pain (bone metastasis pain, visceral metastasis pain, etc.).

Next to pain education, clinicians should focus on tackling insomnia and stress, which might improve CS [79]. There is evidence of treatment for insomnia and stress in cancer survivors. Cognitive behavior therapy (CBT) for insomnia (CBT-I) is the gold standard and treatment for insomnia [93]. Systematic review and meta-analysis have shown that CBT-I is strongly recommended for treating insomnia [94]. Cognitive behavioral stress management, which allows patients to better deal with the impact of the environment, had a positive effect on stress in patients with breast cancer [95]. Mindfulness-based stress reduction (MBSR) and yoga are also effective for stress in cancer survivors [96,97,98]. However, evidence is lacking concerning the impact of those interventions on cancer survivors’ pain (cancer pain and pain after cancer treatment) and CS symptoms. The indirect effect of those interventions on CS symptoms should be further investigated in the future. 

## 10. Future Directions for Research and Clinical Practice

Previous clinical studies have examined the efficacy of certain treatments for certain diseases and have not individually designed treatments for problems at the individual patient level. Precision medicine, which has been the focus of much attention in recent years, involves the classification of patients into subgroups based on a multifaceted evaluation of disease and the implementation of treatment tailored to the characteristics of each patient. Precision medicine is mainly used in oncology to identify the histology and genotype of cancer and optimize treatment in individual patients [99]. Precision rehabilitation has not been fully explored at this time. For precision rehabilitation, physical, cognitive, and psychosocial factors need to be measured, and the patients could be classified into subgroups based on results. Some studies classified patients based on CSI scores (low-CSI/high-CSI). High-intensity training improves symptoms of CS in patients with chronic low back pain, and this effect is greatest in those with high CSI scores at baseline. PNE is more effective in pain catastrophizing in patients with high CSI scores [100]. These studies indicate the possibility of developing precision rehabilitation, while there is still a lack of suggestions on how to deal specifically with CSS. Interventions targeting the following sub-categories (1. Emotional distress, 2. Urological and general symptoms, 3. Headache/Jaw symptoms, 4. Sleep disturbance, and 5. Muscle symptoms of CSS) may be needed (Figure 1). Furthermore, it is necessary to determine whether precision cancer pain medicine, customized according to the underlying pain mechanisms, is more effective than conventional medical care. Precision pain medicine should shift from local therapies like stretching, resistance training, and physical therapy to systemic therapies like pain education and activity level pacing (Figure 2). However, the effectiveness of precision pain medicine or precision rehabilitation is not clear for both musculoskeletal patients and cancer survivors, and further research is needed.

## 11. Conclusions

Evidence that CS affects cancer pain is accumulating. Recently, a seven-step diagnostic approach for differentiating the predominant pains has been developed for cancer survivors. Besides, bedside QSTs and CSI-tool could also help clinicians identify CSS. In the future, it is necessary to investigate multimodal lifestyle interventions in the long term for cancer survivors with predominant CS.

## Figures and Tables

**Figure 1 jcm-12-04606-f001:**
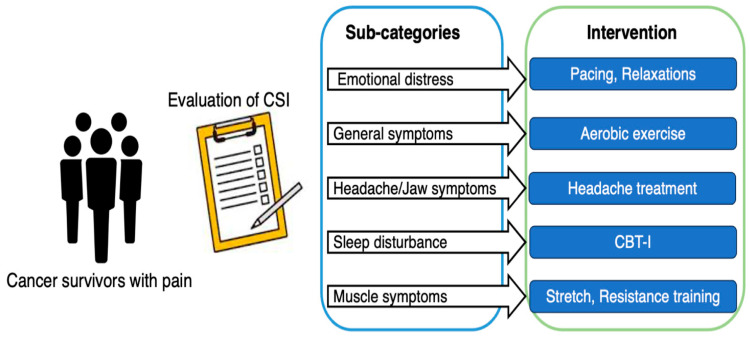
Central sensitivity syndrome targeted education.

**Figure 2 jcm-12-04606-f002:**
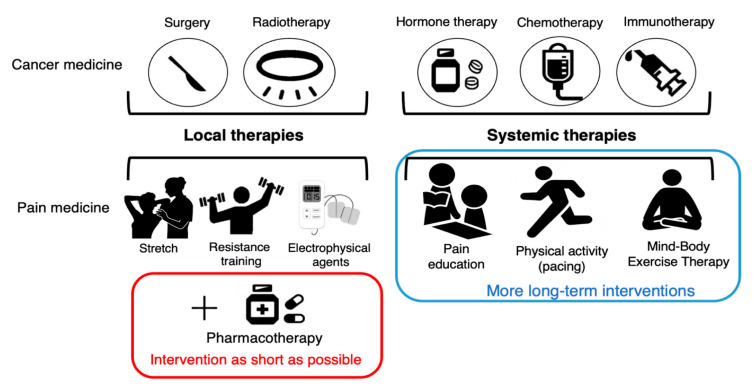
Multimodal therapy for cancer pain.

**Table 1 jcm-12-04606-t001:** Difference between pain neuroscience education for musculoskeletal pain and cancer pain.

Sessions	Musculoskeletal Pain	Cancer Pain
Pathoanatomic models	No reference to pathoanatomic models	Explanation of general side effects of treatment modalities (surgery, radiotherapy, chemotherapy and hormone therapy)
Acute pain vs. chronic pain	Transition from acute to chronic pain	
Nerve function	The neuron (receptor, axon, terminal) and the synapse (action potential, neurotransmitters, postsynaptic membrane potential, chemically driven ion channel)	
Peripheral and central sensitization	Peripheral sensitization (e.g., peripheral nerve injury, inflammation) Central sensitization (e.g., brain and spinal cord function, the pain matrix in the brain)	
Descending nociceptive inhibition and facilitation	Emotions, stress, sleep, physical activity, pain cognitions, and pain behavior	
Reconceptualization of pain as a normal brain response to perceived threat	Vicious cycle due to kinesiophobia and fear–avoidance models. Information about sensations that are a threat to the body is recognized as ‘pain’ as a normal response of the brain.	Vicious cycle due to kinesiophobia and fear–avoidance models. Threatening perception of pain as cancer recurrence or metastasis, leading to heightened anxiety and avoidance behavior.
Transfer knowledge about pain to an adaptive behavioral change	Advice on managing factors that contribute to persistent pain, such as correcting misperceptions and activity management (pacing strategies), taking into account biopsychosocial factors	

## Data Availability

Not applicable.

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
