# Peer review of "Central Sensitization in Cancer Survivors and Its Clinical Implications: State of the Art"

_jcm, 2023, doi:10.3390/jcm12144606_

Round 1

Reviewer 1 Report

I agree with you that we need to "investigate mutimodal lifestyle interventions in the long term for cancer survivors" and these interventions must include, since diagnosis, nutritional aspects, physical activity and sleep management in a more holistic model of care. In this integrative approach, a role can be played also by non pharmacological treatments, such as mindfulness and acupuncture, which you only mention, without specific references- I’d suggest this one:

Effectiveness of Electroacupuncture or Auricular Acupuncture vs Usual Care for Chronic Musculoskeletal Pain Among Cancer Survivors: The PEACE Randomized Clinical Trial - PubMed (nih.gov)

I don’t find easy to understand (and perhaps useful) the Table 1

Moderate English editing is required

Author Response

Comment 1

I agree with you that we need to "investigate mutimodal lifestyle interventions in the long term for cancer survivors" and these interventions must include, since diagnosis, nutritional aspects, physical activity and sleep management in a more holistic model of care. In this integrative approach, a role can be played also by non pharmacological treatments, such as mindfulness and acupuncture, which you only mention, without specific references- I’d suggest this one:

Effectiveness of Electroacupuncture or Auricular Acupuncture vs Usual Care for Chronic Musculoskeletal Pain Among Cancer Survivors: The PEACE Randomized Clinical Trial - PubMed (nih.gov)

Response:

Thanks, great suggestion. We have added the reference.

Comment 2

I don’t find easy to understand (and perhaps useful) the Table 1

Response:

Page 7: We have corrected the Table 1 and adjusted the title of the table.

Comment 3

Moderate English editing is required

Response:

Thank you for this comment. We have corrected our manuscript throughout.

Reviewer 2 Report

The manuscript: “Central Sensitization in Cancer Survivors and its clinical implications: State-of-the-Art” is dedicated to the very important topic of treating pain in patients after cancer treatment with special attention to the central sensitization that is developing in the CNS due to prolong intensive nociceptive input. The review is based on a comprehensive search of existing meta-analyses, obtained clinical and preclinical data from the literature, and the findings of the authors of this review. Based on analyzed data authors indicated directions for future investigation as well as the treatment of CS in cancer survivors. All of these make the manuscript very important and interesting for clinicians and preclinical scientists. The review is based on a solid number of published data, very well-written, and structured. However, a few changes should be made to improve the quality of the manuscript. 

Main concerns

My biggest concern is that despite the review being about pain in cancer survivors it is based on meta-analyses of musculoskeletal pain, pain in cancer patients, and pain after cancer treatment. However, musculoskeletal pain is not covering all cases of cancer growth. Also, cancer pain and pain after cancer treatment have differences in mechanisms and processing. Therefore, to clarify conclusions and clinical recommendations for treatments of CS in cancer survivors these limitations and distinctions must be clearly mentioned in the text of all subsections.

 Line #208 9. Challenges of treating pain in cancer survivors - Targeting central sensitization. Line #223 Authors made not fully correct conclusion. The authors mentioned that:"A systematic review revealed that physical therapy such as manual therapy, exercise, electrotherapy, education, and acupuncture improved CS-related variables in patients with chronic musculoskeletal pain [74]". However, in the cited work effects of the physical therapies on cancer pain or pain in cancer survivors have not been studied because all mentioned forms of musculoskeletal pain did not have a neoplastic origin. Moreover, the authors of the cited work found that only manual therapy, among the other methods, induced a significant effect on CS. Therefore, the authors’ conciliation that "The result indicates the possibility of improving cancer pain via nonpharmacological treatment for CS" needs better evidence. Moreover, analyses of other studies that have been made later in the text [75-80] indicate that currently effects of physical therapies alone on cancer pain are not fully accepted. Also, it is not clear could these methods of treatment be useful for cancer survivors. 

Line # 162 “Moreover, microglia play a significant role in maintaining advanced stage cancer pain in women by generating the inflammatory cytokine IL-1β and increasing synaptic transmission of spinal nociceptive neurons [56].”

It should be mentioned in the text that this data was obtained in preclinical experiments with female rats but not in women.

Minor concerns:

Abstract, line # 29, The abbreviation PNE first appeared in the abstract and should be translated. It should be good to put the abbreviation in parentheses just after “Pain neuroscience education”.

Line # 150. “6. Inflammation and central sensitization in cancer survivors” must be in bold.

Author Response

Major concern 1

My biggest concern is that despite the review being about pain in cancer survivors it is based on meta-analyses of musculoskeletal pain, pain in cancer patients, and pain after cancer treatment. However, musculoskeletal pain is not covering all cases of cancer growth. Also, cancer pain and pain after cancer treatment have differences in mechanisms and processing. Therefore, to clarify conclusions and clinical recommendations for treatments of CS in cancer survivors these limitations and distinctions must be clearly mentioned in the text of all subsections.

Response:

Thank you for insightful comments. We have rewritten it clearly divided into cancer pain, and pain after cancer treatment as your suggestion. First, with reference to the classification of cancer survivor pain in ICD-11 [11], cancer pain is caused by damage of primary cancer, metastasis (e.g. bone pain or visceral metastasis pain), or cancer treatments, and these treatments can induce chronic secondary pain syndromes that persists after cancer treatment such as postmastectomy pain or post-thoracotomy pain after surgery, chemotherapy-induced peripheral neuropathy, aromatase inhibitor-induced musculoskeletal symptoms, radiation-induced neuropathy or radiation-induced fibrosis. This paper distinguishes 1) pain related to cancer or during its treatment and 2) persistent pain after treatment completion (except for maintenance therapy).

Page 2-3, line 95-102: “With reference to the classification of cancer survivor pain in ICD-11 [11], cancer pain is caused by damage of primary cancer, metastasis (e.g. bone pain or visceral metastasis pain), or cancer treatments, and these treatments can induce chronic secondary pain syndromes that persists after cancer treatment such as postmastectomy pain or post-thoracotomy pain after surgery, chemotherapy-induced peripheral neuropathy, aromatase inhibitor-induced musculoskeletal symptoms, radiation-induced neuropathy or radiation-induced fibrosis. This paper distinguishes 1) pain related to the cancer or during its treatment and 2) persistent pain after treatment completion (except for maintenance therapy).

Page 3, line 137-142: “Scott et al. reported that radiotherapy for bone metastatic pain improves hypersensitivity at the pain site [44]. However, there are no reports on hypersensitivity at distant area from the pain site, and the relationship between cancer pain (e.g. bone metastatic pain and visceral metastatic pain) and CS is not clear. Thus, further research is needed to determine whether the relationship between post- cancer treatment pain and CS in cancer survivors is similar for cancer pain.”

Page 4, line 168-174: “The association between pain after cancer treatment and CS-related symptoms in cancer survivors is clear, but the association with cancer pain (e.g. bone and visceral metastases) is still unclear. Cancer survivors with advanced cancer pain and those receiving palliative care or opioid treatment generally have more CS-related symptoms, such as insomnia and fatigue [55-58]. Thus, assessment and intervention for CS-related symptoms will be important for cancer survivors with cancer pain in the future as part of cancer pain management.”

Reference

  1. Scott, AC.; McConnell, S.; Laird, B.; Colvin, L.; Fallon, M. Quantitative Sensory Testing to assess the sensory characteristics of cancer-induced bone pain after radiotherapy and potential clinical biomarkers of response. European Journal of Pain. 2012, 16, 123–133.
  2. Henson, LA.; Maddocks, M.; Evans, C.; Davidson, M.; Hicks, S.; Higginson, IJ. Palliative Care and the Management of Common Distressing Symptoms in Advanced Cancer: Pain, Breathlessness, Nausea and Vomiting, and Fatigue. J Clin Oncol. 2020, 38, 905-914.
  3. Nzwalo, I.; Aboim, MA.; Joaquim, N.; Marreiros, A.; Nzwalo, H. Systematic Review of the Prevalence, Predictors, and Treatment of Insomnia in Palliative Care. Am J Hosp Palliat Care. 2020, 37, 957-969.
  4. Fabi, A.; Bhargava, R.; Fatigoni, S.; Guglielmo, M.; Horneber, M.; Roila, F.; Weis, J.; Jordan, K.; Ripamonti, CI. Cancer-related fatigue: ESMO Clinical Practice Guidelines for diagnosis and treatment. Ann Oncol. 2020, 31, 713-723.
  5. Wang, XS.; Zhao, F.; Fisch, MJ.; O'Mara, AM.; Cella, D.; Mendoza, TR.; Cleeland, CS. Prevalence and characteristics of moderate to severe fatigue: a multicenter study in cancer patients and survivors. Cancer. 2014, 120, 425-32.

Major concern 2

Line #208 9. Challenges of treating pain in cancer survivors - Targeting central sensitization. Line #223 Authors made not fully correct conclusion. The authors mentioned that:"A systematic review revealed that physical therapy such as manual therapy, exercise, electrotherapy, education, and acupuncture improved CS-related variables in patients with chronic musculoskeletal pain [74]". However, in the cited work effects of the physical therapies on cancer pain or pain in cancer survivors have not been studied because all mentioned forms of musculoskeletal pain did not have a neoplastic origin. Moreover, the authors of the cited work found that only manual therapy, among the other methods, induced a significant effect on CS. Therefore, the authors’ conciliation that "The result indicates the possibility of improving cancer pain via nonpharmacological treatment for CS" needs better evidence. Moreover, analyses of other studies that have been made later in the text [75-80] indicate that currently effects of physical therapies alone on cancer pain are not fully accepted. Also, it is not clear could these methods of treatment be useful for cancer survivors.

Response:

Thank you for this comment. We have added and corrected the sentences.

Page 6, line 252-256: “A systematic review revealed that physical therapy results in a modest improvement in CS variables such as TS and CPM in in patients with chronic musculoskeletal pain. It is not clear whether physical therapy improve CS variables in patients with cancer pain and pain after cancer treatment because the systematic review did not include them”

Major concern 3

Line # 162 “Moreover, microglia play a significant role in maintaining advanced stage cancer pain in women by generating the inflammatory cytokine IL-1β and increasing synaptic transmission of spinal nociceptive neurons [56].” It should be mentioned in the text that this data was obtained in preclinical experiments with female rats but not in women.

Response:

We apologize for the inaccurate description. We have added and corrected the sentences.

Page 4, line 1888-196: Moreover, microglia play a significant role in maintaining advanced stage cancer pain in female rats by generating the inflammatory cytokine IL-1β and increasing synaptic transmission of spinal nociceptive neurons [62]. Despite the likelihood of inflammation's involvement in CS among cancer survivors, there is currently only support of preclinical experiments and there is a lack of studies evaluating inflammatory markers in this population and investigating their association with CS. This remains an important area for future research, which will enhance our understanding of how to tackle inflammation for cancer pain and post-treatment pain in this cancer survivor population.

Minor concern 1

Abstract, line # 29, The abbreviation PNE first appeared in the abstract and should be translated. It should be good to put the abbreviation in parentheses just after “Pain neuroscience education”.

Response

We apologize for the inaccurate description. We have added the words (page 1, line 29).

Minor concern 2

Line # 150. “6. Inflammation and central sensitization in cancer survivors” must be in bold.

Response

We have corrected the sentences (page 4, line 176).

Round 2

Reviewer 2 Report

The authors significantly improved this review manuscript, in my opinion. However, I have a small concern that the authors could consider without a need for a re-review.

Line 68. “…CS-related symptoms predict predictors of persistent post-surgical pain intensity…”. I think it will be good to edit the text with “predict predictors”.

Author Response

We have corrected from "CS-related symptoms predict
predictors of persistent post‑surgical pain intensity and interference
at 1 year postoperatively" to "CS-related symptoms have the capability to predict the intensity and interference of persistent post-surgical pain 1 year after surgery".